# Reconstruction of Mercury's internal magnetic field beyond the octupole

Simon Toepfer[1], Ida Oertel[1], Vanita Schiron[1], Yasuhito Narita[2,3], Karl-Heinz Glassmeier[3,4],
Daniel Heyner[3], Patrick Kolhey[3], and Uwe Motschmann[1,5]

[1]Institut für Theoretische Physik, Technische Universität Braunschweig, Braunschweig, Germany
[2]Space Research Institute, Austrian Academy of Sciences, Graz, Austria
[3]Institut für Geophysik und extraterrestrische Physik, Technische Universität Braunschweig, Braunschweig, Germany
[4]Max-Planck-Institut für Sonnensystemforschung, Göttingen, Germany
[5]DLR Institute of Planetary Research, Berlin, Germany

**Correspondence:** Simon Toepfer (s.toepfer@tu-braunschweig.de)

**Abstract.** The reconstruction of Mercury's internal magnetic field enables us to take a look into the inner heart of Mercury. In view of the BepiColombo mission, Mercury's magnetosphere is simulated using a hybrid plasma code and the multipoles of the internal magnetic field are estimated from the virtual spacecraft data using three distinct reconstruction methods: the truncated singular value decomposition, the Tikhonov regularization and Capon's minimum variance projection. The study shows that a precise determination of Mercury's internal field beyond the octupole, up to the dotriacontapole is possible and that Capon's method provides the same high performance as the Tikhonov regularization, which is superior to the performance of the truncated singular value decomposition.

## 1 Introduction

The in-depth analysis of planetary magnetic fields is a key element to understand the structure and dynamics of planetary interiors (Glassmeier and Heyner, 2021). In particular, revealing the nature of Mercury's markedly weak internal magnetic field is one of the primary goals of the BepiColombo mission (Benkhoff et al., 2010, 2021). Thereby, the detailed reconstruction of Mercury's internal field from the total measured magnetic field is of major importance for modelling Mercury's internal dynamo process. Especially, the determination of higher orders of Mercury's internal multipole field reduces the degrees of freedom within the dynamo models (e.g. Heyner et al. (2021)). Due to the plasma physical interaction with the solar wind, the magnetic field around Mercury is contributed not only by the planetary internal field (such as the dynamo-generated field and the crustal remanent field) but also by the external component resulting from currents flowing within the magnetosphere (e.g. Glassmeier and Heyner (2021); Wang et al. (2021)). For the reconstruction of Mercury's internal magnetic field, each part of the field has to be parametrized properly.

Besides the parametrization of the magnetic field, the application of a robust inversion method for separating the magnetic field contributions is required. Several inversion methods have successfully been applied to the reconstruction of planetary

magnetic fields. For example, the Tikhonov regularization (Tikhonov et al., 1995), also known as $L^2$-regularization has been proposed for the analysis of Mercury's magnetic field. Katsura et al. (2021) separate the internal and external magnetic field contributions by making use of the Tikhonov regularization for estimating Mercury's inner core size. Wardinski et al. (2019) and Wardinski et al. (2021) use a weighted Tikhonov regularization for reconstructing Mercury's internal magnetic field from MESSENGER data and also estimate the size of Mercury's core. On the other hand, Connerney (1981) and Connerney et al. (2018) successfully reconstructed Jupiter's magnetic field by using a generalized inverse based on the singular value decomposition. Wang et al. (2021) also use the singular value decomposition for reconstructing Mercury's magnetospheric field. Furthermore, the Capon method (Capon, 1969) is currently being considered as a robust inversion method for Mercury's magnetic field analysis (Toepfer et al., 2020a, b). From the first point of view it is unknown which inversion method provides more reliable results for the wanted model coefficients and thus, the existing inversion methods have to be compared and calibrated. For example, Toepfer et al. (2020a) analyzed the performance of Capon's estimator in comparision with the least square fit estimator and showed that Capon's method provides more accurate results than the least square fit method.

As mentioned in Wardinski et al. (2021), former analyses of Mercury's magnetic field are based on potential field methods for the internal and external magnetic field contributions (Oliveira et al., 2015; Thébault et al., 2018; Wardinski et al., 2019) or on analyzing magnetic equator crossings of the spacecraft (Anderson et al., 2012). For a detailed review of internal field models of Mercury, the reader is referred to the paper of Heyner et al. (2021). The analysis of the MESSENGER data provided an axisymmetric internal dipole field with a dipole moment of $-190\,\mathrm{nT}\,R_{\mathrm{M}}^3$ which is shifted northward by $0.2\,R_{\mathrm{M}}$, where $R_{\mathrm{M}} = 2440\,\mathrm{km}$ denotes the radius of Mercury (Anderson et al., 2012). This field can equivalently be described as a superposition of multipole fields with the internal Gauss coefficients $g_1^0 \approx -190\,\mathrm{nT}$ for the dipole field, $g_2^0 \approx -78\,\mathrm{nT}$ for the quadrupole field, $g_3^0 \approx -20\,\mathrm{nT}$ for the octupole field, $g_4^0 \approx -6\,\mathrm{nT}$ for the hexadecapole field and $g_5^0 \approx -2\,\mathrm{nT}$ for the dotriacontapole field (Anderson et al., 2012; Thébault et al., 2018; Wardinski et al., 2019). The resulting coefficients are slightly varying for different analyzed data sets of the whole MESSENGER mission (e.g., data from different altitude ranges). Plattner and Johnson (2021) and Wardinski et al. (2021) detected magnetic anomalies and nonaxisymmetric internal magnetic field contributions in the northern hemisphere. However, due to the geometry of the orbits, the MESSENGER mission only provided detailed information about the field in the northern hemisphere so that the reconstructed coefficients are correlated and covarying. The symmetrical distribution of the planned Mercury Planetary Orbiter (MPO) trajectories enables a more objective analysis of the internal magnetic field (e.g. Heyner et al. (2021)).

In preparation for the analysis of the BepiColombo magnetic field data the goal of the present study is the comparison of Capon's method with the truncated singular value decomposition and the Tikhonov regularization in application to simulated Mercury magnetic field data. Although the simulated data have to be regarded as a proxy for the not yet available MPO data, the great advantage of the application to simulated data lies in the fact that the exact solution of the inversion problem is known from the input of the simulation which enables the comparison of the estimators with the ideal solution. The plasma interaction of Mercury with the solar wind is simulated using a hybrid plasma code. The resulting magnetic field data are parametrized

by making use of a combination of the Gauss representation (Gauss, 1839; Glassmeier and Tsurutani, 2014) with the Mie representation (Backus, 1986, 1996; Olsen, 1997), called the Gauss-Mie representation, that has successfully been applied to reconstruct Mercury's internal magnetic field up to the octupole term (third degree in the multipole expansion) (Toepfer et al., 2021a). In the current study, the parametrization is extended up to Mercury's internal hexadecapole and dotriacontapole contributions, i.e. degrees 4 and 5. Afterwards, the truncated singular value decomposition, the Tikhonov regularization as well as Capon's method are compared and the inversion methods are applied to the simulated magnetic field data for reconstructing Mercury's internal field up to the fifth degree.

## 2 The Gauss-Mie representation

The fluxgate magnetometer on board the Mercury Planetary Orbiter (MPO) (Glassmeier et al., 2010; Heyner et al., 2021) is built to measure the magnetic field $\boldsymbol{B}$ around Mercury along elliptic orbits. These orbits are conceptually covered by a spherical shell with inner radius $a$, outer radius $c$ and mean radius $b = (a+c)/2$. Since the shell in general includes current-carrying regions with $\partial_{\boldsymbol{x}} \times \boldsymbol{B} \neq 0$, the most commonly used Gauss representation (Gauss, 1839; Glassmeier and Tsurutani, 2014) does not yield a proper characterization of Mercury's magnetospheric magnetic field. By virtue of the superposition principle, the

total measured field $\boldsymbol{B}$ can be decomposed into $\boldsymbol{B}^i$, $\boldsymbol{B}^e$ and $\boldsymbol{B}^{sh}$. $\boldsymbol{B}^i$ is the irrotational field contribution internally generated by currents flowing beneath the shell ($r < a$) and $\boldsymbol{B}^e$ is the irrotational field contribution externally generated by currents flowing above the shell ($r > c$). Shortly, we call $\boldsymbol{B}^i$ the internal field and $\boldsymbol{B}^e$ the external field. $\boldsymbol{B}^{sh}$ is the rotational field generated by currents flowing within the shell. Considering the Mie representation (Backus, 1986, 1996; Olsen, 1997; Toepfer et al., 2021a) the currents flowing within the shell in the region $a < r < c$ generate toroidal $\boldsymbol{B}^{sh}_T$ and poloidal $\boldsymbol{B}^{sh}_P$ magnetic

field structures that superpose with the existing internal and external fields, delivering

$$\boldsymbol{B} = \boldsymbol{B}^i + \boldsymbol{B}^e + \boldsymbol{B}^{sh}_T + \boldsymbol{B}^{sh}_P, \tag{1}$$

which is called the Gauss-Mie representation of the magnetic field (Toepfer et al., 2021a).

Since both the internal and external fields $\boldsymbol{B}^i$ and $\boldsymbol{B}^e$ are purely poloidal and irrotational, these parts can be parametrized

via the Gauss representation. Therefore, there exist scalar potentials $\Phi^i$ and $\Phi^e$ with

$$\boldsymbol{B}^i = -\partial_{\boldsymbol{x}} \Phi^i \tag{2}$$

and

$$\boldsymbol{B}^e = -\partial_{\boldsymbol{x}} \Phi^e. \tag{3}$$

Using the body-fixed, planetary-centered spherical coordinates with radius $r \in [R_{\mathrm{M}}, \infty)$, azimuth angle $\lambda \in [0, 2\pi]$ and co-

latitude $\theta \in [0, \pi]$, the scalar potentials can be expanded into spherical harmonics, yielding

$$\Phi^i = R_{\mathrm{M}} \sum_{l=1}^{\infty} \sum_{m=0}^{l} \left(\frac{R_{\mathrm{M}}}{r}\right)^{l+1} [g_l^m \cos(m\lambda) + h_l^m \sin(m\lambda)] P_l^m(\cos(\theta)) \tag{4}$$

and

$$\Phi^e = R_{\mathrm{M}} \sum_{l=1}^{\infty} \sum_{m=0}^{l} \left(\frac{r}{R_{\mathrm{M}}}\right)^{l} [q_l^m \cos(m\lambda) + s_l^m \sin(m\lambda)] P_l^m(\cos(\theta)), \tag{5}$$

where $R_{\mathrm{M}}$ is the planetary radius of Mercury and $P_l^m$ are the Schmidt-normalized associated Legendre polynomials of degree

$l$ and order $m$ (Abramowitz and Stegun, 1972). Spherical harmonic expansion of the scalar potentials was introduced by the epoch-making work by Gauss (1839), also revisited by Glassmeier and Tsurutani (2014) with the contemporary English translation. The potential $\Phi^i$ is determined by the internal Gauss coefficients $g_l^m$ and $h_l^m$, whereas the external Gauss coefficients

$q_l^m$ and $s_l^m$ describe the external field contributions.

In general, the shell includes current-carrying regions and therefore, the contributions $\boldsymbol{B}_T^{sh}$ and $\boldsymbol{B}_P^{sh}$ cannot be described by the Gauss representation. Making use of the Mie representation, one may introduce scalar functions $\Psi_T^{sh}$ and $\Psi_P^{sh}$ in the spirit of potential functions by defining the magnetic field through the curl of the scalar functions (multiplied by the position vector) as follows:

$$\boldsymbol{B}_T^{sh} = \partial_{\boldsymbol{x}} \times \left( \Psi_T^{sh} \boldsymbol{r} \right) \tag{6}$$

and

$$\boldsymbol{B}_P^{sh} = \partial_{\boldsymbol{x}} \times \left[ \partial_{\boldsymbol{x}} \times \left( \Psi_P^{sh} \boldsymbol{r} \right) \right], \tag{7}$$

where $\boldsymbol{r} = r\,\boldsymbol{e}_r$ and $\boldsymbol{e}_r$ is the unit vector in radial direction. Because of the underlying spherical geometry, it is straightforward to expand the scalar functions $\Psi_T^{sh}$ and $\Psi_P^{sh}$ into spherical harmonics. In contrast to the scalar potentials $\Phi^i$ and $\Phi^e$, the exact radial dependence of the corresponding expansion coefficients in the Mie representation is unknown. Thus, it is useful to

perform a special Taylor series expansion for the coefficients with respect to the radius $r$ in the vicinity of the mean radius $b$ of the spherical shell (Toepfer et al., 2021a), providing

$$\Psi_T^{sh} = \frac{R_{\mathrm{M}}}{r} \sum_{l=1}^{\infty} \sum_{m=0}^{l} \left[ \alpha_l^m + \alpha_l'^m \rho + \mathcal{O}(\rho^2) \right] P_l^m \left( \cos(\theta) \right) \tag{8}$$

and

$$\Psi_P^{sh} = \frac{R_{\mathrm{M}}^2}{r} \sum_{l=1}^{\infty} \sum_{m=0}^{l} \left[ \beta_l^m + \beta_l'^m \rho + \mathcal{O}(\rho^2) \right] P_l^m \left( \cos(\theta) \right), \tag{9}$$

where the variable $\rho = (r - b)/R_{\mathrm{M}}$ denotes the signed radial distance from the mean shell radius $b$ in units of the planetary radius and the Big-O-notation $\mathcal{O}$ summarizes the orders of the Taylor series expansion higher than (or equal to) $\rho^2$. The expansion coefficients are given by

$$\alpha_l^m = a_l^m \cos(m\lambda) + b_l^m \sin(m\lambda),$$
$$\alpha_l'^m = a_l'^m \cos(m\lambda) + b_l'^m \sin(m\lambda),$$
$$\beta_l^m = c_l^m \cos(m\lambda) + d_l^m \sin(m\lambda),$$
$$\beta_l'^m = c_l'^m \cos(m\lambda) + d_l'^m \sin(m\lambda).$$

Due to the geometry of the MPO orbits the application of the thin shell approximation (Backus, 1986, 1996; Olsen, 1997; Toepfer et al., 2021a) is a valid assumption. Whereas the locally generated poloidal field $\boldsymbol{B}_P^{sh}$ cannot be distinguished from the internal and external contributions $\boldsymbol{B}^i$ and $\boldsymbol{B}^e$ within the reconstruction procedure (Toepfer et al., 2021a), the thin shell

approximation enables us to separate the poloidal field into its internal and external parts. When the shell thickness is smaller

than the length scale on which the currents change in radial direction, the poloidal field $\boldsymbol{B}_P^{sh}$ generated by toroidal currents flowing within the shell can be neglegted compared to the internal $\boldsymbol{B}^i$ and external $\boldsymbol{B}^e$ poloidal fields. Combining the Gauss representation with the Mie representation by making use of the thin shell approximation, the magnetic field can be rewritten in the linear algebraic form

$$\boldsymbol{B} = -\partial_{\boldsymbol{x}}\Phi^i - \partial_{\boldsymbol{x}}\Phi^e + \partial_{\boldsymbol{x}} \times \left(\Psi_T^{sh}\boldsymbol{r}\right) = \mathbf{H}\boldsymbol{g}, \tag{10}$$

where the terms of the series expansions are arranged into the shape matrix $\mathbf{H}$ which solely contains known information about the measurement positions. The corresponding expansion coefficients describing the amplitude of the field are summarized into the vector $\boldsymbol{g}$.

It should be noted that the internal field is canonically described in a Mercury-Body-Fixed corotating coordinate system (MBF), whereas the external field is canonically described in a Mercury-Solar-Orbital system (MSO) with the $x$-axis orientated towards the sun, the $z$-axis orientated parallel to the rotation axis, i.e. antiparallel to the internal dipole moment, and the $y$-axis completes the right-handed system (Toepfer et al., 2021a; Heyner et al., 2021). In the present study, simulated Mercury magnetic field data are evaluated. Therefore, the internal and external fields are expressed in a Mercury-Anit-Solar-Orbital coordinate system (MASO), where the $x$-axis is orientated towards the nightside of Mercury (away from the sun), the $z$-axis is orientated parallel to the rotation axis and the $y$-axis completes the right-handed system (Toepfer et al., 2021a).

In general, the number of data points from an orbital mission is much larger than the number of wanted model coefficients. Thus, the resulting shape matrix $\mathbf{H}$ is rectangular and its inverse $\mathbf{H}^{-1}$ does not exist (Toepfer et al., 2020b). Therefore, Eq. (10) describes an overdetermined system of linear equations. A unique solution for the desired model coefficients $\boldsymbol{g}$ is not available, so the coefficients have to be estimated from the measurements by making use of a suitable inversion method.

## 3 Inversion methods

For the reconstruction of Mercury's internal magnetic field, various kinds of data inversion techniques are available such as the least square fit method, the singular value decomposition, the Tikhonov regularization and Capon's minimum variance method (e.g., Haykin, 2014). The construction of these inversion techniques is reviewed along with merits and demerits in this section.

### 3.1 Least square fit

The most prominent inversion method for linear inversion problems is the least square fit method (LSF) (e.g., Haykin, 2014). The method minimizes the quadratic deviation

$$\text{minimize} \quad \left| \mathbf{H} \boldsymbol{g} - \boldsymbol{B} \right|^2 \tag{11}$$

between the model $\mathbf{H}\boldsymbol{g}$ and the measurements $\boldsymbol{B}$ with respect to the set of model parameters $\boldsymbol{g}$ with unknown values, resulting in the least square fit estimator

$$\boldsymbol{g}_L = \left[ \mathbf{H}^\dagger \mathbf{H} \right]^{-1} \mathbf{H}^\dagger \boldsymbol{B}, \tag{12}$$

where the dagger $\dagger$ denotes the Hermitian conjugate. Thus, the LSF method solely weights the data by the shape matrix $\mathbf{H}$. When the measurements are completely described by the underlying model, the LSF estimator provides the most accurate estimation for the wanted model coefficients (Toepfer et al., 2020a).

### 3.2 Singular value decomposition

The singular value decomposition (SVD) generalizes the LSF method (e.g., Haykin, 2014). The method is based on the decomposition of the shape matrix $\mathbf{H} \in \mathbb{R}^{m \times n}$ with rank $n$ into the form

$$\mathbf{H} = \mathbf{U} \boldsymbol{\Sigma} \mathbf{V}^\dagger, \tag{13}$$

where $\mathbf{U} \in \mathbb{R}^{m \times m}$ and $\mathbf{V} \in \mathbb{R}^{n \times n}$ are orthonormal transformations, so that $\mathbf{U}^\dagger = \mathbf{U}^{-1}$ as well as $\mathbf{V}^\dagger = \mathbf{V}^{-1}$ is valid (e.g., Connerney (1981); Haykin (2014); Connerney et al. (2018)). The matrix

$$\boldsymbol{\Sigma} = \begin{bmatrix} \Sigma_1 & \dots & 0 \\ \vdots & \ddots & \vdots \\ 0 & \dots & \Sigma_n \\ 0 & \dots & 0 \\ \vdots & \ddots & \vdots \\ 0 & \dots & 0 \end{bmatrix} \in \mathbb{R}^{m \times n}, \tag{14}$$

where $\Sigma_1 \geq \cdots \geq \Sigma_n \geq 0$ (sorted in descending order), contains the so-called singular values $\Sigma_i$ of the shape matrix $\mathbf{H}$, which are defined as the square-roots of the eigenvalues of the matrix $\mathbf{H}^\dagger \mathbf{H}$. Inserting the singular value decomposition of the matrix $\mathbf{H}$ (Eq. 13) into Eq. (10) delivers the singular value estimator

$$\boldsymbol{g}_S = \mathbf{H}^+ \boldsymbol{B} = \mathbf{V} \boldsymbol{\Sigma}^+ \mathbf{U}^\dagger \boldsymbol{B} = \sum_{i=1}^{n} \frac{1}{\Sigma_i} \boldsymbol{v}_i \boldsymbol{u}_i^\dagger \boldsymbol{B} \tag{15}$$

for the wanted model coefficients $\boldsymbol{g}$, where the vectors $\boldsymbol{u}_i$ and $\boldsymbol{v}_i$ are the corresponding column vectors of the matrices $\mathbf{U}$ and $\mathbf{V}$, respectively (e.g., Haykin, 2014). The matrix

$$\mathbf{H}^+ = \mathbf{V} \boldsymbol{\Sigma}^+ \mathbf{U}^\dagger = \sum_{i=1}^{n} \frac{1}{\Sigma_i} \boldsymbol{v}_i \boldsymbol{u}_i^\dagger \tag{16}$$

is called the "pseudo-inverse" or generalized inverse of the matrix $\mathbf{H}$, where the matrix $\boldsymbol{\Sigma}^+$ is given by

$$\left(\boldsymbol{\Sigma}^+\right)_{ij} = \begin{cases} \frac{1}{\Sigma_i} & , \quad i = j \\ 0 & , \quad \text{else} \end{cases} . \tag{17}$$

In the case of a full rank, the matrix $\mathbf{H}^\dagger \mathbf{H}$ is invertible and thus, the estimator resulting from the singular value decomposition restores the least square fit estimator

$$\boldsymbol{g}_S = \mathbf{H}^+ \boldsymbol{B} = \left[\mathbf{H}^\dagger \mathbf{H}\right]^{-1} \mathbf{H}^\dagger \boldsymbol{B} = \boldsymbol{g}_L. \tag{18}$$

Conferring to Eq. (15), the solution $\boldsymbol{g}_S$ (as well as the solution $\boldsymbol{g}_L$) is determined by the inverse of singular values. In the case of small or even vanishing singular values, i.e., $\text{rank}(\mathbf{H}) < n$, the solution diverges. Furthermore, the condition number

$$\kappa(\mathbf{H}) = \frac{\max\limits_i \Sigma_i}{\min\limits_i \Sigma_i} = \frac{\Sigma_1}{\Sigma_n} \tag{19}$$

of the shape matrix increases $\kappa(\mathbf{H}) \gg 1$ for small singular values and thus, the inversion problem (Eq. 10) is said to be ill-posed. A large condition number impairs the solvability of the inversion problem and furthermore increases the error propagation (Toepfer et al., 2021b).

One of the most commonly used techniques for reducing the condition number is the low rank approximation or truncated singular value decomposition (TSVD) (Eckart and Young, 1936). Within this approximation only $k$ singular values, where $k < n$, are considered within the solution and therefore, the shape matrix $\mathbf{H}$ is approximated by a shape matrix

$$\mathbf{H}_k = \mathbf{U}_k \boldsymbol{\Sigma}_k \mathbf{V}_k^\dagger, \tag{20}$$

where the matrices $\mathbf{U}_k$ and $\mathbf{V}_k$ are composed of the first $k$ column vectors of the matrices $\mathbf{U}$ and $\mathbf{V}$, respectively. The diagonal elements of the matrix $\boldsymbol{\Sigma}_k$ are given by the largest $k$ singular values of the shape matrix $\mathbf{H}$. Therefore, the matrix $\mathbf{H}_k$ has a lower rank, i.e., $\text{rank}(\mathbf{H}_k) = k$, and a lower condition number

$$\kappa(\mathbf{H}_k) = \frac{\Sigma_1}{\Sigma_k} < \frac{\Sigma_1}{\Sigma_n} = \kappa(\mathbf{H}) \tag{21}$$

is achieved. Due to the modification of the shape matrix, the solution $\boldsymbol{g}_S$ transfers onto

$$\boldsymbol{g}_{TSVD} = \mathbf{H}_k^+ \boldsymbol{B} = \sum_{i=1}^{k} \frac{1}{\Sigma_i} \boldsymbol{v}_i \boldsymbol{u}_i^\dagger \boldsymbol{B}, \tag{22}$$

which will be called the TSVD estimator.

    On one hand, the decreased condition number improves the solvability of the inversion method. On the other hand, it should

be noted that the elimination of singular values causes a lack of information which can decrease the performance of the data analysis. This lack of information can be quantitatively estimated via the model resolution matrix (Menke, 2012; Heyner et al., 2021)

$$\mathbf{R} = \mathbf{H}_k^+ \mathbf{H}. \tag{23}$$

Because of $\boldsymbol{g}_{TSVD} = \mathbf{R}\boldsymbol{g}$, where $\boldsymbol{g}$ denotes the ideal (imaginary) solution for the model coefficients, the model resolution

matrix identifies the resolution of each model coefficient. In the case of $\mathbf{R} \neq \mathbf{I}$, the estimated coefficients are determined by a linear combination of the ideal solution and thus, there exist model covariances. If $\mathbf{R} = \mathbf{I}$, each coefficient is resolved for 100% (Menke, 2012). To achieve the highest performance of the estimator, the condition number and the lack of information have to come to a compromise (Connerney, 1981; Toepfer et al., 2021a).

### 3.3    Tikhonov regularization

Within the application of the LSF method and the singular value decomposition, the wanted model coefficients are required to satisfy the highly overdetermined system of linear equations $\boldsymbol{B} = \mathbf{H}\boldsymbol{g}$. From a physical point of view an additional constraint for the solution (for example minimum energy) can be incorporated to reduce the degrees of freedom within the estimation.

    Considering the analysis of Mercury's internal magnetic field, the wanted model coefficients describe the amplitude of the

magnetic field. Thus, it is obvious that the currents flowing within the magnetosphere as well as inside of Mercury generate magnetic fields with a minimal energy. The energy spectrum $W_l$ of each spherical harmonic degree $l$ of the internal magnetic field at Mercury's surface is represented by the Mauersberger-Lowes spectrum (Mauersberger, 1956; Lowes, 1966)

$$W_l = (l+1) \sum_{m=0}^{l} \left[ (g_l^m)^2 + (h_l^m)^2 \right]. \tag{24}$$

Therefore, minimizing the energy spectrum $W_l$ is equivalent to the minimization of the norm of the wanted model coefficients

$\left| \boldsymbol{g} \right|^2$ and thus it is useful to search for solutions having the minimal norm. This constraint is known as the Tikhonov regularization, $L^2$-regularization or minimum norm solution (e.g., Tikhonov et al. (1995); Haykin (2014)). Using this additional constraint, the original least square fit problem (cf. Eq. 11) can be extended as

$$\text{minimize} \quad \left| \mathbf{H}\boldsymbol{g} - \boldsymbol{B} \right|^2 + \alpha \left| \boldsymbol{g} \right|^2, \tag{25}$$

where $\alpha$ is the so-called regularization parameter which describes the corresponding Lagrange multiplier of the additional constraint. The Tikhonov estimator for the wanted model coefficients results in

$$\boldsymbol{g}_T = \left[\mathbf{H}^\dagger\mathbf{H} + \alpha\mathbf{I}\right]^{-1}\mathbf{H}^\dagger\boldsymbol{B}, \tag{26}$$

where $\mathbf{I}$ denotes the identity matrix. Thus, the additional constraint modifies the shape matrix and therefore, modifies the location dependence of the measurement points with respect to the underlying model.

Inserting the singuar value decomposition of the matrix $\mathbf{H}$ into Eq. (26), delivers

$$
\begin{aligned}
\boldsymbol{g}_T &= \left[\mathbf{H}^\dagger\mathbf{H} + \alpha\mathbf{I}\right]^{-1}\mathbf{H}^\dagger\boldsymbol{B} \\
&= \left[\left(\mathbf{U}\boldsymbol{\Sigma}\mathbf{V}^\dagger\right)^\dagger\mathbf{U}\boldsymbol{\Sigma}\mathbf{V}^\dagger + \alpha\mathbf{I}\right]^{-1}\left(\mathbf{U}\boldsymbol{\Sigma}\mathbf{V}^\dagger\right)^\dagger\boldsymbol{B} \\
&= \left[\mathbf{V}\boldsymbol{\Sigma}^\dagger\mathbf{U}^\dagger\mathbf{U}\boldsymbol{\Sigma}\mathbf{V}^\dagger + \alpha\mathbf{I}\right]^{-1}\mathbf{V}\boldsymbol{\Sigma}^\dagger\mathbf{U}^\dagger\boldsymbol{B} \\
&= \left[\mathbf{V}\boldsymbol{\Sigma}^\dagger\boldsymbol{\Sigma}\mathbf{V}^\dagger + \alpha\mathbf{V}\mathbf{V}^\dagger\right]^{-1}\mathbf{V}\boldsymbol{\Sigma}^\dagger\mathbf{U}^\dagger\boldsymbol{B} \\
&= \left[\mathbf{V}\left(\boldsymbol{\Sigma}^\dagger\boldsymbol{\Sigma} + \alpha\mathbf{I}\right)\mathbf{V}^\dagger\right]^{-1}\mathbf{V}\boldsymbol{\Sigma}^\dagger\mathbf{U}^\dagger\boldsymbol{B} \\
&= \mathbf{V}\left[\boldsymbol{\Sigma}^\dagger\boldsymbol{\Sigma} + \alpha\mathbf{I}\right]^{-1}\mathbf{V}^\dagger\mathbf{V}\boldsymbol{\Sigma}^\dagger\mathbf{U}^\dagger\boldsymbol{B} \\
&= \mathbf{V}\left[\boldsymbol{\Sigma}^\dagger\boldsymbol{\Sigma} + \alpha\mathbf{I}\right]^{-1}\boldsymbol{\Sigma}^\dagger\mathbf{U}^\dagger\boldsymbol{B} \\
&= \mathbf{V}\boldsymbol{\Sigma}_T^+\mathbf{U}^\dagger\boldsymbol{B} \\
&= \mathbf{H}_T^+\boldsymbol{B} \tag{27}
\end{aligned}
$$

where the matrix

$$\mathbf{H}_T^+ = \mathbf{V}\boldsymbol{\Sigma}_T^+\mathbf{U}^\dagger \tag{28}$$

will be called the Tikhonov inverse and

$$\boldsymbol{\Sigma}_T^+ = \left[\boldsymbol{\Sigma}^\dagger\boldsymbol{\Sigma} + \alpha\mathbf{I}\right]^{-1}\boldsymbol{\Sigma}^\dagger = \begin{bmatrix} \frac{\Sigma_1}{\Sigma_1^2+\alpha} & 0 & \cdots & 0 & 0 & \cdots & 0 \\ 0 & \frac{\Sigma_2}{\Sigma_2^2+\alpha} & \ddots & \vdots & 0 & \cdots & 0 \\ \vdots & \ddots & \ddots & 0 & \vdots & \ddots & \vdots \\ 0 & \cdots & 0 & \frac{\Sigma_n}{\Sigma_n^2+\alpha} & 0 & \cdots & 0 \end{bmatrix} \in \mathbb{R}^{n\times m}. \tag{29}$$

Thus, the original shape matrix $\mathbf{H}$ transfers onto the shape matrix

$$\mathbf{H}_T = \mathbf{U}\boldsymbol{\Sigma}_T\mathbf{V}^\dagger, \tag{30}$$

where

$$\boldsymbol{\Sigma}_T = \begin{bmatrix} \frac{\Sigma_1^2+\alpha}{\Sigma_1} & 0 & \cdots & 0 \\ 0 & \frac{\Sigma_2^2+\alpha}{\Sigma_2} & \ddots & \vdots \\ 0 & 0 & \ddots & 0 \\ 0 & \cdots & 0 & \frac{\Sigma_n^2+\alpha}{\Sigma_n} \\ 0 & \cdots & 0 & 0 \\ \vdots & \ddots & \vdots & \vdots \\ 0 & \cdots & 0 & 0 \end{bmatrix} \in \mathbb{R}^{m \times n}, \tag{31}$$

which has a lower condition number

$$
\begin{aligned}
\kappa(\mathbf{H}_T) = \kappa\left(\mathbf{H}_T^+\right) &= \frac{\max\limits_{i} \frac{\Sigma_i^2+\alpha}{\Sigma_i}}{\min\limits_{i} \frac{\Sigma_i^2+\alpha}{\Sigma_i}} \\
&= \frac{\max\limits_{i}\left(\Sigma_i + \frac{\alpha}{\Sigma_i}\right)}{\min\limits_{i}\left(\Sigma_i + \frac{\alpha}{\Sigma_i}\right)} \\
&\leq \frac{\max\limits_{i}\Sigma_i + \alpha\max\limits_{i}\frac{1}{\Sigma_i}}{\min\limits_{i}\Sigma_i + \alpha\min\limits_{i}\frac{1}{\Sigma_i}} \\
&= \frac{\Sigma_1 + \frac{\alpha}{\Sigma_n}}{\Sigma_n + \frac{\alpha}{\Sigma_1}} \\
&= \frac{\Sigma_1\Sigma_n + \alpha}{\Sigma_n} \frac{\Sigma_1}{\Sigma_1\Sigma_n + \alpha} \\
&= \frac{\Sigma_1}{\Sigma_n} = \kappa(\mathbf{H}).
\end{aligned}
\tag{32}
$$

Therefore, the modification of the shape matrix improves the solvability of the inversion problem in analogy to the TSVD.

Comparison of the SVD estimator

$$\boldsymbol{g}_S = \sum_{i=1}^{n} \frac{1}{\Sigma_i} \boldsymbol{v}_i \boldsymbol{u}_i^\dagger \boldsymbol{B} \tag{33}$$

with the Tikhonov estimator

$$\boldsymbol{g}_T = \mathbf{H}_T^+ \boldsymbol{B} = \sum_{i=1}^{n} \frac{\Sigma_i}{\Sigma_i^2+\alpha} \boldsymbol{v}_i \boldsymbol{u}_i^\dagger \boldsymbol{B} \tag{34}$$

shows that the original singular values are modified by $\alpha$. In the case of small or even vanishing singular values, the solution $\boldsymbol{g}_T$ remains finite. Furthermore, within the Tikhonov solution, $n$ (modified) singular values are considered, whereas within the

TSVD solution (Eq. 22) only $k < n$ singular values are incorporated. Making use of the TSVD, the information that is included within the truncated singular values is eliminated. Within the application of the Tikhonov regularization, small singular values are not truncated, but shifted. Thus, the information which is contained in the shifted singular values is still present in a modified form. In the case of $\alpha = 0$, the Tikhonov estimator transfers onto the LSF estimator.

In analogy to the TSVD, the modification of the shape matrix results in a non-trivial resolution matrix

$$\mathbf{R}_T = \mathbf{H}_T^+ \mathbf{H} \neq \mathbf{I} \tag{35}$$

and thus, in general there exist model covariances.

### 3.4 Capon's method

Capon's minimum variance projection is broadly established in the analysis of seismic and plasma waves (Capon, 1969; Motschmann et al., 1996; Narita, 2012). In view of the BepiColombo mission, the method is currently being considered as a robust inversion method for the analysis of Mercury's internal magnetic field (Toepfer et al., 2020a, b, 2021a).

Due to the complexity of Mercury's magnetosphere, the entire parametrization of the magnetic field contributions, generated by currents flowing within the magnetosphere is unrealizable. Thus, it is useful to decompose the magnetic field $\boldsymbol{B}$ into parametrized parts $\mathbf{H}\boldsymbol{g}$ (cf. Eq. 10), non-parametrized parts $\boldsymbol{v}$ as well as measurement noise $\boldsymbol{n}$, so that

$$\boldsymbol{B} = \mathbf{H}\boldsymbol{g} + \boldsymbol{v} + \boldsymbol{n} \tag{36}$$

is valid. The measurement noise is assumed to be Gaussian with variance $\sigma_n^2$ and zero mean ($\langle \boldsymbol{n} \rangle = 0$). Since the shape matrix $\mathbf{H}$ is not invertible and the non-parametrized parts are unknown, the exact solution for the wanted model coefficients $\boldsymbol{g}$ is not available in general. Capon's method delivers an estimator $\boldsymbol{g}_C$ for the ideal solution $\boldsymbol{g}$. The method is based on the construction of a filter matrix $\mathbf{w}$, that minimizes the output power

$$P = \text{tr}\left[\mathbf{w}^\dagger \mathbf{M} \mathbf{w}\right] \tag{37}$$

with respect to the distortionless constraint (also referred to as the unit gain constraint)

$$\mathbf{w}^\dagger \mathbf{H} = \mathbf{I}, \tag{38}$$

where $\text{tr}\left[\mathbf{w}^\dagger \mathbf{M} \mathbf{w}\right]$ is the trace of the matrix $\mathbf{w}^\dagger \mathbf{M} \mathbf{w}$ and $\mathbf{I}$ is the identity matrix. The matrix $\mathbf{M} = \langle \boldsymbol{B} \circ \boldsymbol{B} \rangle$ denotes the data covariance matrix. Capon's estimator realizing the minimal output power results in

$$\boldsymbol{g}_C = \mathbf{w}^\dagger \langle \boldsymbol{B} \rangle = \left[\mathbf{H}^\dagger \mathbf{M}^{-1} \mathbf{H}\right]^{-1} \mathbf{H}^\dagger \mathbf{M}^{-1} \langle \boldsymbol{B} \rangle. \tag{39}$$

The comparison of Capon's estimator with the LSF estimator (cf. Eq. 12) shows that Capon's method can be interpreted as a special case of the weighted least square fit method (Toepfer et al., 2020b). The robustness of the method can be improved by

the diagonal loading technique $\mathbf{M} \to \mathbf{M} + \sigma_d^2 \mathbf{I}$, where $\sigma_d$ is the diagonal loading parameter (Toepfer et al., 2020b). Inserting the diagonally loaded data covariance matrix $\mathbf{M} = \langle \boldsymbol{B} \rangle \circ \langle \boldsymbol{B} \rangle + \sigma^2 \mathbf{I}$, where $\sigma^2 = \sigma_n^2 + \sigma_d^2$, into the output power and making use of $\boldsymbol{g}_C = \mathbf{w}^\dagger \langle \boldsymbol{B} \rangle$, delivers

$$
\begin{aligned}
P = \mathrm{tr} \left[ \mathbf{w}^\dagger \mathbf{M} \mathbf{w} \right] &= \mathrm{tr} \left[ \mathbf{w}^\dagger \langle \boldsymbol{B} \rangle \circ \langle \boldsymbol{B} \rangle \mathbf{w} \right] + \sigma^2 \, \mathrm{tr} \left[ \mathbf{w}^\dagger \mathbf{w} \right] \\
&= \mathrm{tr} \left[ \boldsymbol{g}_C \circ \boldsymbol{g}_C \right] + \sigma^2 \, \mathrm{tr} \left[ \mathbf{w}^\dagger \mathbf{w} \right] \\
&= \left| \boldsymbol{g}_C \right|^2 + \sigma^2 \, \mathrm{tr} \left[ \mathbf{w}^\dagger \mathbf{w} \right].
\end{aligned}
\tag{40}
$$

Thus, Capon's method minimizes the energy spectrum in analogy to the Tikhonov regularization. In contrast to the Tikhonov regularization, Capon's method additionally weights the data by the inverse data covariance matrix (cf. Eq. 39).

In the case of small singular values of the shape matrix $\mathbf{H}$, the performance of Capon's method decreases in analogy to the LSF method. Since Capon's method already minimizes the energy spectrum $P$, the method cannot be improved by making use of the Tikhonov regularization and thus, the TSVD should be applied to reduce the condition number. Within many applications it is unknown, which singular values have to be considered within the solution and which singular values can be dropped. On the other hand, the application of the Tikhonov regularization modifies the condition number of the original shape matrix ($\kappa \left( \mathbf{H}_T \right) \leq \kappa \left( \mathbf{H} \right)$) and thus, the Tikhonov regularization delivers the "optimal" condition number for solving the problem. Therefore, the number of singular values that have to be considered within the solution can be estimated by choosing $k$, so that

$$
\kappa \left( \mathbf{H}_k \right) \approx \kappa \left( \mathbf{H}_T \right)
\tag{41}
$$

is valid. Due to the application of the TSVD, Capon's filter matrix is modified to

$$
\mathbf{w}_k^\dagger = \left[ \mathbf{H}_k^\dagger \mathbf{M}^{-1} \mathbf{H}_k \right]^{-1} \mathbf{H}_k^\dagger \mathbf{M}^{-1},
\tag{42}
$$

on the cost of violating the unit gain condition (Toepfer et al., 2021a), i.e.

$$
\mathbf{R}_C = \mathbf{w}_k^\dagger \mathbf{H} \neq \mathbf{I}.
\tag{43}
$$

## 4 Application to simulated Mercury magnetic field data

In the following, the above presented inversion methods are applied to simulated Mercury magnetic field data for reconstructing Mercury's internal multipole field up to the dotriacontapole term. The internal field is modeled as being generated by the dynamo field (crustal fields are not considered here) and the multipole spectrum model is taken from the MESSENGER results (Anderson et al., 2012; Thébault et al., 2018; Wardinski et al., 2019). The magnetic field is sampled along the trajectories of virtual spacecraft (representing the MPO spacecraft) and the multipole spectrum is estimated using the different inversion techniques for the virtual spacecraft data. The estimators for the spectrum are compared to the internal Gauss coefficients implemented in the simulation.

### 4.1 Hybrid simulation of Mercury's magnetosphere

The magnetic field resulting from the plasma interaction of Mercury with the solar wind is simulated with the hybrid code AIKEF (Müller et al., 2011), that has successfully been applied to several problems in Mercury's plasma interaction (e.g., Müller et al. (2011); Exner et al. (2018, 2020)). The internal Gauss coefficients $g_1^0 = -190\,\mathrm{nT}$ (dipole field), $g_2^0 = -78\,\mathrm{nT}$ (quadrupole field), $g_3^0 = -20\,\mathrm{nT}$ (octupole field), $g_4^0 = -6\,\mathrm{nT}$ (hexadecapole field) and $g_5^0 = 8\,\mathrm{nT}$ (dotriacontapole field) (Anderson et al., 2012; Thébault et al., 2018; Wardinski et al., 2019) are implemented in the simulation code. For the first validation, the value $g_5^0 = 8\,\mathrm{nT}$ was chosen from a synthetic Mercury magnetic field model of Thébault et al. (2018). The interplanetary magnetic field with a magnitude of $B_{\mathrm{IMF}} = 20\,\mathrm{nT}$ is orientated along the vector $(x, y, z)^T = (0, 0, 1)^T$ in the Mercury-Anti-Solar-Orbital (MASO) frame. The magnetic field data are simulated under stationary solar wind conditions, where the solar wind velocity of $v_{\mathrm{sw}} = 400\,\mathrm{km/s}$ points along the $x$-axis (Milillo et al., 2020). The solar wind proton density number is chosen to $n_{\mathrm{sw}} = 30\,\mathrm{cm}^{-3}$ and the electron $T_e$ and proton temperatures $T_p$ are implemented to $T_e = T_p = 2.5 \times 10^5\,\mathrm{K}$ (Exner et al., 2020; Milillo et al., 2020). The resulting magnitude of the magnetic field in the $x$-$z$-plane is displayed in Figure 1.

The geometry of Mercury's magnetosphere is mainly dominated by the internal dipole field. Also the internal quadrupole field in terms of the apparently northward shifted dipole field is visible. The influence of the octupole, hexadecapole and dotriacontapole fields is not visually noticeable within the figure.

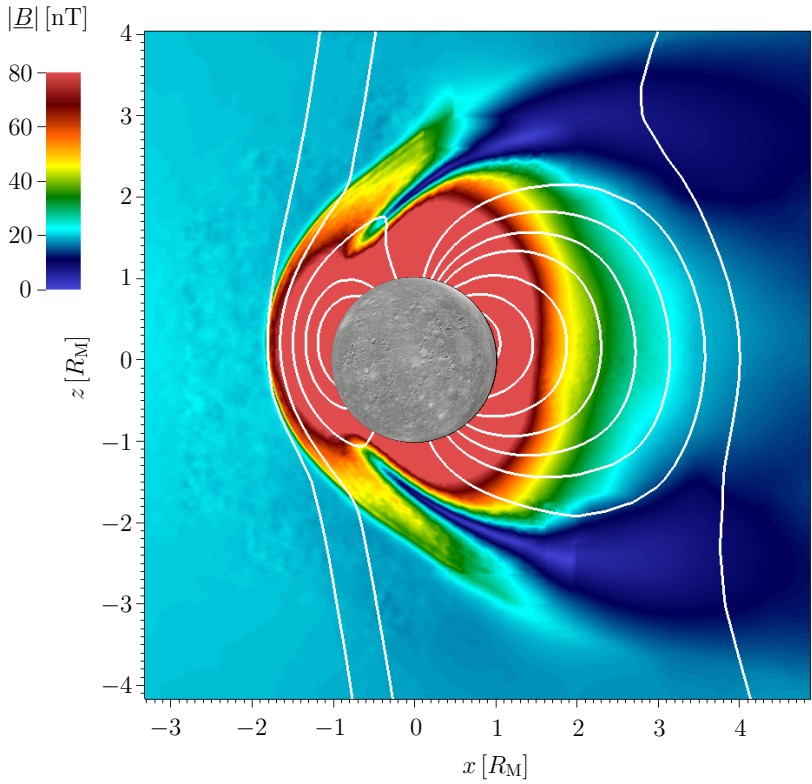

**Figure 1.** Simulated magnitude of the magnetic field $\boldsymbol{B}$ in the $x$-$z$-plane where internal multipoles are implemented from dipole to dotria-contapole. The white lines represent the magnetic field lines and the grey circle of radius $1\,R_{\mathrm{M}}$ symbolizes Mercury.

## 4.2 Reconstruction of the multipole coefficients

The internal Gauss coefficients implemented in the simulation code represent the ideal solution $\boldsymbol{g}$ for the data inversion. For the reconstruction of the coefficients, the magnetic field data are evaluated along meridional elliptic orbits around Mercury, representing the planned MPO orbits. The orbital plane is rotated about the rotation axis ($z$-axis) from $-50°$ (afternoon/post-midnight sector) over $0°$ (noon/midnight, $x$-$z$-plane) to $50°$ (morning/pre-midnight sector). Figure 2 displays the distribution of the synthetically generated measurement positions in the $x$-$z$-plane. The internal and external potentials ($\Phi^i$, $\Phi^e$) are expanded into spherical harmonics up the fifth degree, representing the internal/external dipole, quadrupole, octupole, hexadecapole and dotriacontapole field. The terms of the degrees $l \leq 4$ are expanded up to the order $m = l$. The dotriacontapole ($l = 5$) is only expanded up to the zeroth order ($m = 0$) for simplicity. The scalar function $\Psi_T^{sh}$ of the toroidal magnetic field $\boldsymbol{B}_T^{sh}$ is expanded into spherical harmonics up to the second degree and order and additionally into a first order Taylor series for the radial distance. The scalar function $\Psi_P^{sh}$ of the poloidal magnetic field $\boldsymbol{B}_P^{sh}$ is neglected by making use of the thin shell approximation. Thus, the magnetic field is described by 66 expansion coefficients with 25 internal Gauss coefficients, 25 external Gauss coefficients and 16 toroidal coefficients. These coefficients are to be determined from the data by the inversion techniques. Since the exact solution for the internal Gauss coefficients is a priori known as the inputs to the simulation, in the following discussion we will focus on the reconstructed 25 internal coefficients.

The optimal regularization parameter for the application of the Tikhonov regularization results in $\alpha \approx 0.9$, so that $\kappa\left(\mathbf{H}_T\right) \approx 86$. Incorporating all the 66 singular values of the shape matrix results in a condition number of $\kappa\left(\mathbf{H}\right) \approx 540$ for the original shape matrix. Considering 60 singular values within the reconstruction procedure, the condition number decreases to $\kappa\left(\mathbf{H}_{60}\right) \approx 90$. Thus, for the calculation of Capon's estimator and for the TSVD estimator 60 singular values are incorporated. The optimal diagonal loading parameter for the application of Capon's method results in $\sigma = 590\,\mathrm{nT}$. The optimal regularization parameter $\alpha$ as well as the optimal diagonal loading parameter $\sigma$ can be determined by minimizing the deviation between the corresponding estimator and the ideal solution, which is known from the input of the simulation. Within the practical application of the methods in future satellite experiments, the exact solution is unknown and thus, the regularization parameter as well as the diagonal loading parameter can be estimated by making use of the $L$-curve technique (Toepfer et al., 2020b). The estimators resulting from the TSVD method, Capon's method and the Tikhonov regularization are displayed in Table 1.

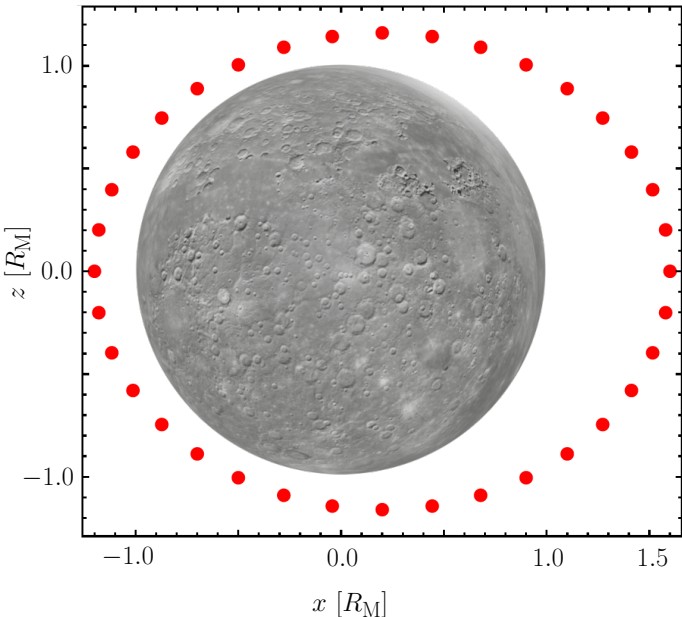

**Figure 2.** Synthetically generated measurement positions (red dots) in the $x$-$z$-plane ($\lambda = 0°$). The orbital plane is rotated about the rotation axis ($z$-axis) from $-50°$ over $0°$ to $50°$. The grey cirle of radius $1\,R_{\mathrm{M}}$ symbolizes Mercury.

**Table 1.** Implemented and reconstructed internal Gauss coefficients for the dipole, quadrupole, octupole, hexadecapole and dotriacontapole field. The implemented value $g_5^0$ of the dotriacontapole field was chosen from a synthetic Mercury magnetic field model of Thébault et al. (2018). The errors are derived by synthetically disturbing the simulated data and measurement positions as discussed in the text.

| Coefficient | Input in nT | TSVD in nT | Capon in nT | Tikhonov in nT |
|:---:|:---:|:---:|:---:|:---:|
| $g_1^0$ | -190.0 | $-197.0 \pm 0.3$ | $-191.4 \pm 0.1$ | $-192.2 \pm 0.3$ |
| $g_1^1$ | 0.0 | $-1.3 \pm 0.2$ | $-1.2 \pm 0.2$ | $-1.9 \pm 0.2$ |
| $h_1^1$ | 0.0 | $-0.3 \pm 1.0$ | $-0.3 \pm 1.0$ | $-0.2 \pm 0.9$ |
| $g_2^0$ | -78.0 | $-76.9 \pm 0.3$ | $-74.7 \pm 0.4$ | $-76.4 \pm 0.3$ |
| $g_2^1$ | 0.0 | $2.6 \pm 0.2$ | $2.5 \pm 0.2$ | $2.8 \pm 0.1$ |
| $h_2^1$ | 0.0 | $0.0 \pm 0.3$ | $0.0 \pm 0.2$ | $0.1 \pm 0.2$ |
| $g_2^2$ | 0.0 | $0.2 \pm 0.2$ | $0.2 \pm 0.2$ | $-0.2 \pm 0.2$ |
| $h_2^2$ | 0.0 | $0.1 \pm 0.1$ | $0.1 \pm 0.1$ | $0.1 \pm 0.1$ |
| $g_3^0$ | -20.0 | $-19.9 \pm 0.2$ | $-19.3 \pm 0.2$ | $-21.0 \pm 0.2$ |
| $g_3^1$ | 0.0 | $-1.0 \pm 0.0$ | $-1.0 \pm 0.0$ | $-1.3 \pm 0.1$ |
| $h_3^1$ | 0.0 | $0.4 \pm 0.2$ | $0.4 \pm 0.2$ | $0.3 \pm 0.2$ |
| $g_3^2$ | 0.0 | $-0.4 \pm 0.2$ | $-0.3 \pm 0.2$ | $-2.5 \pm 0.2$ |
| $h_3^2$ | 0.0 | $0.1 \pm 0.2$ | $0.1 \pm 0.2$ | $0.1 \pm 0.2$ |
| $g_3^3$ | 0.0 | $0.5 \pm 0.2$ | $0.4 \pm 0.2$ | $0.3 \pm 0.1$ |
| $h_3^3$ | 0.0 | $0.0 \pm 1.0$ | $0.0 \pm 1.0$ | $0.0 \pm 1.0$ |
| $g_4^0$ | -6.0 | $-4.3 \pm 0.2$ | $-4.2 \pm 0.2$ | $-5.2 \pm 0.2$ |
| $g_4^1$ | 0.0 | $0.3 \pm 0.1$ | $0.3 \pm 0.1$ | $0.3 \pm 0.0$ |
| $h_4^1$ | 0.0 | $-0.1 \pm 0.2$ | $0.0 \pm 0.2$ | $0.0 \pm 0.2$ |
| $g_4^2$ | 0.0 | $0.4 \pm 0.0$ | $0.4 \pm 0.0$ | $0.0 \pm 0.0$ |
| $h_4^2$ | 0.0 | $0.1 \pm 0.3$ | $0.1 \pm 0.3$ | $0.1 \pm 0.3$ |
| $g_4^3$ | 0.0 | $1.2 \pm 0.2$ | $1.2 \pm 0.2$ | $-0.2 \pm 0.2$ |
| $h_4^3$ | 0.0 | $-0.1 \pm 0.5$ | $-0.1 \pm 0.5$ | $-0.1 \pm 0.5$ |
| $g_4^4$ | 0.0 | $0.0 \pm 0.2$ | $0.0 \pm 0.2$ | $0.0 \pm 0.2$ |
| $h_4^4$ | 0.0 | $-0.1 \pm 0.7$ | $-0.1 \pm 0.7$ | $-0.1 \pm 0.6$ |
| $g_5^0$ | 8.0 | $7.2 \pm 0.0$ | $7.0 \pm 0.0$ | $7.1 \pm 0.0$ |

The deviations between the reconstructed and the implemented coefficients result in $\left|\boldsymbol{g}_{TSVD} - \boldsymbol{g}\right|/\left|\boldsymbol{g}\right| \approx 3.9\%$, $\left|\boldsymbol{g}_C - \boldsymbol{g}\right|/\left|\boldsymbol{g}\right| \approx 2.6\%$ and $\left|\boldsymbol{g}_T - \boldsymbol{g}\right|/\left|\boldsymbol{g}\right| \approx 2.6\%$, so that the reconstructed and the implemented coefficients are in good agreement.

Since the TSVD method only weights the data by the position information, the TSVD estimator yields the largest deviation. The deviation of Capon's estimator equals the deviation of the Tikhonov estimator, since both methods incorporate the constraint of a minimum norm solution. Especially, Mercury's internal hexadecapole and dotriacontapole fields can be reconstructed from the data to a good precision. Furthermore, the model resolution matrices of Capon's estimator and the Tikhonov regularization are of the same order and close to the identity matrix

$$\mathbf{R}_T \approx \mathbf{R}_C \approx \mathbf{I}, \tag{44}$$

providing a high model resolution. Thus, the eliminated singular values do not contain any additional information, that may improve the inversion. Taking into account 66 singular values within the calculation of Capon's estimator and the TSVD estimator, the TSVD estimator transfers onto the LSF estimator, so that $\tilde{\mathbf{R}}_C = \mathbf{R} = \mathbf{I}$. In this case, the deviation between Capon's estimator and the ideal solution results in $\left|\boldsymbol{g}_C - \boldsymbol{g}\right|/\left|\boldsymbol{g}\right| \approx 3.3\%$ for $\sigma = 420\,\mathrm{nT}$ and $\left|\boldsymbol{g}_L - \boldsymbol{g}\right|/\left|\boldsymbol{g}\right| \approx 6.2\%$ for the LSF estima-
370 tor. Thus, the errors can be reduced by making use of the TSVD for the shape matrix. Furthermore, it should be noted that the application of the Gauss-Mie representation improves the inversion results significantly in comparison to the sole parametrization of the field via the Gauss representation (Toepfer et al., 2021a). Additionally, the reconstructed Gauss coefficients of the external irrotational field are in agreement with the values reconstructed in former works (Wardinski et al., 2019; Toepfer et al., 2021a).

In the present study, simulated magnetic field data and synthetically generated measurement positions are evaluated. Within the practical application to in-situ spacecraft data, it is expectable that the measurement positions as well as the measurements will be determined defectively, resulting in estimation errors (e.g., Toepfer et al. (2021b)). As a proof of concept, the errors are incorporated by synthetically disturbing the simulated data and the measurement positions. For example, disturbing the data
by a normally distributed error of the width of $1\,\mathrm{nT}$ and zero mean results in a relative error between the disturbed $\tilde{\boldsymbol{B}}$ and the undisturbed data $\boldsymbol{B}$ of $|\tilde{\boldsymbol{B}} - \boldsymbol{B}|/|\boldsymbol{B}| \approx 1\%$. Disturbing the measurement positions by a normally distributed error of the width of $10\,\mathrm{km}$ and zero mean results in a defective shape matrix $\tilde{\mathbf{H}}$ (Toepfer et al., 2021b). The relative error between the defective and the ideal shape matrix $\mathbf{H}$ is given by $||\tilde{\mathbf{H}} - \mathbf{H}||_2/||\mathbf{H}||_2 \approx 3\%$, where $||.||_2$ denotes the spectral norm. The measurement errors and measurement position errors transfer onto disturbed estimators $\tilde{\boldsymbol{g}}_{TSVD}$, $\tilde{\boldsymbol{g}}_C$ and $\tilde{\boldsymbol{g}}_T$ (Toepfer et al., 2021b). The unsigned
difference between the coefficients resulting from the undisturbed data and measurement positions and the coefficients resulting from the disturbed data and measurement positions is used as an estimation for the error of the coefficients listed in Table 1. For all the coefficients together the deviations between the undisturbed and the disturbed estimators result in

$$\frac{\left|\tilde{\boldsymbol{g}}_{TSVD} - \boldsymbol{g}_{TSVD}\right|}{\left|\boldsymbol{g}_{TSVD}\right|} \approx \frac{\left|\tilde{\boldsymbol{g}}_C - \boldsymbol{g}_C\right|}{\left|\boldsymbol{g}_C\right|} \approx \frac{\left|\tilde{\boldsymbol{g}}_T - \boldsymbol{g}_T\right|}{\left|\boldsymbol{g}_T\right|} \approx 1\% \tag{45}$$

so that the inversion methods may be declared as robust. However, the influence of specific measurement errors such as offsets,
gains resulting from thermal variations and spacecraft magnetic disturbances (Narita et al., 2021) on the reconstruction proce-

dure should be analyzed in future works.

The analysis of MESSENGER magnetic field data provided a value of $g_5^0 \approx 2\,\text{nT}$ (Thébault et al., 2018). Thus, the inversion methods are furthermore applied to simulated magnetic field data with the internal Gauss coefficients $g_1^0 = -190\,\text{nT}$ (dipole field), $g_2^0 = -78\,\text{nT}$ (quadrupole field), $g_3^0 = -20\,\text{nT}$ (octupole field), $g_4^0 = -6\,\text{nT}$ (hexadecapole field) and $g_5^0 = 2\,\text{nT}$ (dotriacontapole field). The data are again evaluated along the planned MPO trajectories. The following analyses will be restricted to undisturbed simulated data.

The optimal regularization parameter results in $\alpha \approx 0.77$, so that $\kappa\left(\mathbf{H}_T\right) \approx 93$. Thus, for the calculation of Capon's estimator and for the TSVD estimator again 60 singular values are considered. The optimal diagonal loading parameter results in $\sigma = 670\,\text{nT}$. The estimators of the TSVD method, Capon's method and the Tikhonov regularization are displayed in Table 2.

The relative errors result in $\left|\boldsymbol{g}_{TSVD} - \boldsymbol{g}\right|/\left|\boldsymbol{g}\right| \approx 3.3\%$, $\left|\boldsymbol{g}_C - \boldsymbol{g}\right|/\left|\boldsymbol{g}\right| \approx 2.2\%$ and $\left|\boldsymbol{g}_T - \boldsymbol{g}\right|/\left|\boldsymbol{g}\right| \approx 2.3\%$, so that the results are in agreement with the coefficients presented in Table 1. The performance of Capon's estimator again is as competitive as the performance of the Tikhonov estimator. Due to the smaller value of the internal coefficient $g_5^0$ implemented in the simulation, the relative deviation between the reconstructed and the implemented internal dotriacontapole results in about 50%. Thus, if the true value of Mercury's internal dotriacontapole is of the order of $2\,\text{nT}$, uncertainties within the reconstruction procedure are expectable.

**Table 2.** Implemented and reconstructed internal Gauss coefficients for the dipole, quadrupole, octupole, hexadecapole and dotriacontapole field. The implemented multipole spectrum is taken from the MESSENGER results (Anderson et al., 2012; Thébault et al., 2018; Wardinski et al., 2019).

| Coefficient | Input in nT | TSVD in nT | Capon in nT | Tikhonov in nT |
|---|---|---|---|---|
| $g_1^0$ | -190.0 | -195.5 | -190.9 | -191.5 |
| $g_1^1$ | 0.0 | -1.4 | -1.3 | -2.1 |
| $h_1^1$ | 0.0 | -0.6 | -0.6 | -0.5 |
| $g_2^0$ | -78.0 | -77.6 | -75.7 | -77.5 |
| $g_2^1$ | 0.0 | 2.3 | 2.3 | 2.3 |
| $h_2^1$ | 0.0 | 0.0 | 0.0 | 0.0 |
| $g_2^2$ | 0.0 | 0.3 | 0.2 | -0.3 |
| $h_2^2$ | 0.0 | -0.1 | -0.1 | 0.0 |
| $g_3^0$ | -20.0 | -19.8 | -19.3 | -20.7 |
| $g_3^1$ | 0.0 | -1.0 | -1.0 | -1.3 |
| $h_3^1$ | 0.0 | 0.5 | 0.5 | 0.4 |
| $g_3^2$ | 0.0 | -0.7 | -0.7 | -2.5 |
| $h_3^2$ | 0.0 | 0.1 | 0.1 | 0.1 |
| $g_3^3$ | 0.0 | 0.6 | 0.6 | 0.5 |
| $h_3^3$ | 0.0 | 0.3 | 0.3 | 0.2 |
| $g_4^0$ | -6.0 | -4.4 | -4.3 | -5.1 |
| $g_4^1$ | 0.0 | 0.2 | 0.2 | 0.3 |
| $h_4^1$ | 0.0 | -0.1 | -0.1 | -0.1 |
| $g_4^2$ | 0.0 | 0.5 | 0.5 | 0.1 |
| $h_4^2$ | 0.0 | 0.1 | 0.1 | 0.1 |
| $g_4^3$ | 0.0 | 1.0 | 1.0 | -0.2 |
| $h_4^3$ | 0.0 | 0.0 | 0.0 | 0.0 |
| $g_4^4$ | 0.0 | 0.0 | 0.0 | 0.1 |
| $h_4^4$ | 0.0 | 0.2 | 0.2 | 0.2 |
| $g_5^0$ | 2.0 | 1.3 | 1.2 | 1.2 |

Besides the uncertainty concerning the internal coefficient $g_5^0$, the analysis of the MESSENGER data provided a range for the internal Gauss coefficients of the dipole up to the octupole field (Wardinski et al., 2019). Thus, it is worthwile to analyze the performance of the estimators by evaluating magnetic field data resulting from the lower boundaries of the internal coefficients along the planned MPO trajectories. The inversion methods are applied to (undisturbed) simulated magnetic field data with the internal Gauss coefficients $g_1^0 = -190\,\text{nT}$ (dipole field), $g_2^0 = -57\,\text{nT}$ (quadrupole field), $g_3^0 = -16\,\text{nT}$ (octupole field), $g_4^0 = -4\,\text{nT}$ (hexadecapole field) and $g_5^0 = 8\,\text{nT}$ (dotriacontapole field). The implemented value of the hexadecapole field is chosen arbitrarily and the dotriacontapole field is again taken from the Mercury magnetic field model of Thébault et al. (2018). The optimal regularization parameter results in $\alpha \approx 1.57$, so that $\kappa\left(\mathbf{H}_T\right) \approx 64$ and thus, for the calculation of Capon's estimator and for the TSVD estimator 58 singular values are considered. The optimal diagonal loading parameter results in $\sigma = 515\,\text{nT}$. The estimators of the TSVD method, Capon's method and the Tikhonov regularization are displayed in Table 3, where the relative estimation errors result in $\left|\boldsymbol{g}_{TSVD} - \boldsymbol{g}\right|/\left|\boldsymbol{g}\right| \approx 6.6\%$, $\left|\boldsymbol{g}_C - \boldsymbol{g}\right|/\left|\boldsymbol{g}\right| \approx 3.6\%$ and $\left|\boldsymbol{g}_T - \boldsymbol{g}\right|/\left|\boldsymbol{g}\right| \approx 3.4\%$. In analogy to the examples discussed above, the performance of Capon's estimator again equals the performance of the Tikhonov estimator. Due to the smaller numerical values of the implemented coefficients, the errors are slightly larger than the deviation between the reconstructed and the ideal coefficients presented in Tables 1 and 2, respectively.

Although Mercury's internal magnetic field is dominated by an axisymmetric geometry (Anderson et al., 2012; Wardinski et al., 2019), recent studies indicate the existence of nonaxisymmetric field contributions (Wardinski et al., 2021). Thus, the inversion techniques should be capable of reconstructing nonaxisymmetric internal fields to guarantee an unbiased analysis of the BepiColombo data. Therefore, the inversion methods are furthermore applied to simulated data resulting from arbitrary chosen nonaxisymmetric internal Gauss coefficients with $m \neq 0$. The optimal regularization parameter results in $\alpha \approx 1.39$, so that $\kappa\left(\mathbf{H}_T\right) \approx 68$. Thus, for the calculation of Capon's estimator and for the TSVD estimator again 58 singular values are considered. The optimal diagonal loading parameter results in $\sigma = 560\,\text{nT}$. The estimators of the TSVD method, Capon's method and the Tikhonov regularization as well as the implemented values are displayed in Table 4. The relative estimation errors result in $\left|\boldsymbol{g}_{TSVD} - \boldsymbol{g}\right|/\left|\boldsymbol{g}\right| \approx 6.0\%$, $\left|\boldsymbol{g}_C - \boldsymbol{g}\right|/\left|\boldsymbol{g}\right| \approx 3.7\%$ and $\left|\boldsymbol{g}_T - \boldsymbol{g}\right|/\left|\boldsymbol{g}\right| \approx 3.5\%$. For an objective assessment of the reconstruction quality, the estimation errors of the axisymmetric and the nonaxisymmetric internal field contributions should be discussed separately. The relative deviation of the axisymmtric coefficients with $m = 0$ results in $5.4\%$ for the TSVD estimator, $2.8\%$ for Capon's estimator and $2.0\%$ for the Tikhonov estimator, whereas the relative deviation of the nonaxisymmtric coefficients with $m \neq 0$ results in $34\%$ for the TSVD estimator, $32\%$ for Capon's estimator and $38\%$ for the Tikhonov estimator which is much larger due to the implemented smaller numerical values. Thus, is worthwile to analyze the performance of the estimators for different combinations and numerical values of nonaxisymmetric internal Gauss coefficients in future studies. However, in case mother nature suprises us with nonaxisymmetric internal field contributions at Mercury as suggested by Wardinski et al. (2021), the inversion methods presented here enable the determination of the corresponding internal Gauss coefficients from the MPO data.

**Table 3.** Implemented and reconstructed internal Gauss coefficients for the dipole, quadrupole, octupole, hexadecapole and dotriacontapole field. For the implemented quadrupole and octupole coefficients, the lower boundary from the MESSENGER results (Wardinski et al., 2019) are implemented.

| Coefficient | Input in nT | TSVD in nT | Capon in nT | Tikhonov in nT |
|---|---|---|---|---|
| $g_1^0$ | -190.0 | -201.9 | -191.2 | -192.3 |
| $g_1^1$ | 0.0 | -1.1 | -1.1 | -1.1 |
| $h_1^1$ | 0.0 | -0.3 | -0.3 | -0.4 |
| $g_2^0$ | -57.0 | -56.1 | -53.1 | -54.2 |
| $g_2^1$ | 0.0 | 2.1 | 2.0 | 3.8 |
| $h_2^1$ | 0.0 | -0.4 | -0.4 | -0.2 |
| $g_2^2$ | 0.0 | 0.5 | 0.5 | 1.1 |
| $h_2^2$ | 0.0 | 0.2 | 0.2 | 0.2 |
| $g_3^0$ | -16.0 | -13.6 | -12.9 | -16.5 |
| $g_3^1$ | 0.0 | -0.8 | -0.7 | -0.6 |
| $h_3^1$ | 0.0 | 0.1 | 0.1 | 0.1 |
| $g_3^2$ | 0.0 | 0.8 | 0.8 | -3.2 |
| $h_3^2$ | 0.0 | 0.9 | 0.9 | 1.0 |
| $g_3^3$ | 0.0 | 0.1 | 0.1 | 0.1 |
| $h_3^3$ | 0.0 | 0.1 | 0.1 | 0.2 |
| $g_4^0$ | -4.0 | -2.8 | -2.7 | -4.1 |
| $g_4^1$ | 0.0 | 0.7 | 0.7 | 0.7 |
| $h_4^1$ | 0.0 | -1.0 | -1.0 | -1.0 |
| $g_4^2$ | 0.0 | -0.1 | -0.1 | -1.0 |
| $h_4^2$ | 0.0 | 0.1 | 0.1 | 0.1 |
| $g_4^3$ | 0.0 | 3.8 | 3.6 | 1.2 |
| $h_4^3$ | 0.0 | 0.5 | 0.5 | 0.5 |
| $g_4^4$ | 0.0 | -0.4 | -0.4 | -0.7 |
| $h_4^4$ | 0.0 | -0.3 | -0.3 | -0.2 |
| $g_5^0$ | 8.0 | 7.3 | 6.9 | 7.2 |

**Table 4.** Implemented and reconstructed internal Gauss coefficients for the dipole, quadrupole, octupole, hexadecapole and dotriacontapole field. The implemented axisymmetric coefficients are taken from the MESSENGER results (Anderson et al., 2012; Thébault et al., 2018; Wardinski et al., 2019). For the implemented nonaxisymmetric coefficients, arbitrary values are chosen.

| Coefficient | Input in nT | TSVD in nT | Capon in nT | Tikhonov in nT |
|---|---|---|---|---|
| $g_1^0$ | -190.0 | -201.0 | 192.0 | -192.6 |
| $g_1^1$ | -5.0 | -6.7 | -6.4 | -6.5 |
| $h_1^1$ | 5.0 | 4.2 | 4.0 | 4.4 |
| $g_2^0$ | -78.0 | -77.0 | -73.5 | -75.0 |
| $g_2^1$ | 10.0 | 12.2 | 11.7 | 13.7 |
| $h_2^1$ | 0.0 | -0.3 | -0.3 | -0.1 |
| $g_2^2$ | 0.0 | 0.9 | 0.9 | 1.5 |
| $h_2^2$ | 0.0 | 0.1 | 0.1 | 0.3 |
| $g_3^0$ | -20.0 | -18.4 | -17.6 | -20.9 |
| $g_3^1$ | 10.0 | 8.6 | 8.2 | 8.7 |
| $h_3^1$ | 0.0 | 0.5 | 0.5 | 0.4 |
| $g_3^2$ | 0.0 | 0.8 | 0.8 | -2.7 |
| $h_3^2$ | 0.0 | 0.9 | 0.9 | 1.0 |
| $g_3^3$ | 0.0 | 0.3 | 0.3 | 0.0 |
| $h_3^3$ | 0.0 | 0.5 | 0.5 | 0.3 |
| $g_4^0$ | -6.0 | -4.3 | -4.1 | -5.8 |
| $g_4^1$ | 0.0 | 0.7 | 0.7 | 0.7 |
| $h_4^1$ | 0.0 | -0.8 | -0.8 | -0.8 |
| $g_4^2$ | 0.0 | -0.5 | -0.5 | -1.6 |
| $h_4^2$ | 0.0 | 0.2 | 0.1 | 0.1 |
| $g_4^3$ | 0.0 | 3.7 | 3.5 | 1.4 |
| $h_4^3$ | 0.0 | 0.7 | 0.7 | 0.6 |
| $g_4^4$ | 0.0 | -0.2 | -0.2 | -0.8 |
| $h_4^4$ | 0.0 | 0.3 | 0.3 | 0.0 |
| $g_5^0$ | 8.0 | 7.6 | 7.2 | 7.4 |

As newly established, Capon's method provides the same performance as the Tikhonov regularization. For the comparison of the two methods, a more general comment is appropriate. Both the methods incorporate the constraint of a minimum norm

solution and therefore, deliver superior results than the TSVD method. Within the derivation of the Tikhonov estimator $\boldsymbol{g}_T$, the constraint is included synthetically, whereas the nature of Capon's method is based on the minimization of the output power (Toepfer et al., 2020a, b), which corresponds to the norm of the estimator. Furthermore, Capon's method weights the measurements by the data covariance and the measurement positions. Thus, the weighting of Capon's method is adaptive, since the data determine the weighting by themselves, whereas the Tikhonov method weights all data equally. In view of the

practical application of the inversion methods, both the methods provide nearly comparable results. However, Capon's method incorporates more information from the experiment (position and data) than the Tikhonov regularization.

## 5   Summary and outlook

The detailed characterization of Mercury's internal magnetic field is expected to play an important role in understanding the origin of the field. Due to the interference of the internal parts with the magnetospheric field contributions, on one hand, each

part of the field has to be parametrized properly. On the other hand, a robust inversion method for reconstructing the wanted model coefficients is required.

In preparation for the analysis of the magnetic field measurements provided by the magnetometer on board the MPO, the plasma interaction of Mercury with the solar wind is simulated numerically. The resulting magnetic field data are parametrized

by a combination of the Gauss representation with the Mie representation, called the Gauss-Mie representation and the corresponding expansion coefficients are reconstructed from the data using the truncated singular value decomposition, the Tikhonov regularization as well as Capon's method. The reconstructed internal Gauss coefficients of the dipole, quadrupole, octupole, hexadecapole and dotriacontapole fields are in very good agreement with the coefficients implemented into the simulation code and thus, a high precision determination of Mercury's internal magnetic field up to the dotriacontapole is expectable. The

quality of the reconstructed internal coefficients depends on the magnitude of the values. For example, in the case of an internal dotriacontapole coefficient of $g_5^0 = 8\,\mathrm{nT}$, the reconstructed and the implemented coefficient are in very good agreement. For $g_5^0 = 2\,\mathrm{nT}$ the deviation is of the order of about $1\,\mathrm{nT}$. However, due to the symmetric distribution of the planned MPO orbits around Mercury (Heyner et al., 2021), it is worthwile to consider the reconstruction of higher degrees of the internal field in future studies. Furthermore, the methods presented here are capable of reconstructing potentially existing nonaxisymmetric

magnetic field contributions at Mercury, which enables an objective analysis of the prospective MPO data.

The comparision of the inversion methods shows that Capon's method and the Tikhonov method provide a comparative performance. Since both the methods incorporate the constraint of a minimum norm solution, which is equivalent to minimum energy, Capon's estimator as well as the Tikhonov estimator deliver superior results than the truncated singular value decompo-

sition. It should be noted, that the constraint of a minimum norm solution is included synthetically within the derivation of the

Tikhonov estimator. Since Capon's method is based on the minimization of the output power which corresponds to the norm of the estimator, the constraint of a minimum norm solution is naturally implemented in the method. Furthermore, Capon's method weights the data adaptively, since the weighting is determined by the measurement positions and the measurements themselves, whereas the Tikhonov method weights all data equally. Besides the constraint of a minimum norm solution, further
physically based constraints, for example at the core-mantle boundary (Wardinski et al., 2019) can be incorporated to improve the inversion results (Holme and Bloxham, 1996; Heyner et al., 2021).

Within the analyses presented here, simulated stationary magnetic field data resulting from the plasma interaction of Mercury with the solar wind under constant external conditions, i.e., constant solar wind density, velocity and interplanetary magnetic
field orientation are evaluated. Due to the fast temporal variability of the Hermean environment (e.g., Slavin et al. (2021)) and changing solar wind conditions, a suitable filtering procedure should be applied before analyzing in-situ data. Thereby, data from calm solar wind conditions are preferable. However, in view of the BepiColombo mission (Benkhoff et al., 2010, 2021), the present work provides an overview of the most commonly used inversion methods and shows that Capon's method as well as the Tikhonov method enable a high precision determination of Mercury's internal magnetic field. The subsequent
comparison of the reconstructed internal field resulting from the analysis of the BepiColombo data with the MESSENGER results will open the door to discuss the evidence of yet undetected secular variations at Mercury (e.g. Philpott et al. (2014); Oliveira et al. (2019); Heyner et al. (2021)).

*Data availability.* Simulation data can be provided upon request.

*Author contributions.* All authors contributed conception and design of the study; ST and UM wrote the first draft of the manuscript; All
495 authors contributed to manuscript revision, read and approved the submitted version.

*Funding:* We acknowledge support by the German Research Foundation and the Open Access Publication Funds of the Technische Universität Braunschweig.

The work by Y. Narita is supported by the Austrian Space Applications Programme at the Austrian Research Promotion Agency under contract 865967.

D. Heyner and K.-H. Glassmeier are supported by the German Ministerium für Wirtschaft und Energie and the German Zentrum für Luft- und Raumfahrt under contract 50 QW1501.

*Competing interests.* The authors declare that they have no conflict of interest.

*Acknowledgements.* The authors are grateful for stimulating discussions and helpful suggestions by Alexander Schwenke. ST and UM acknowledge the North-German Supercomputing Alliance (HLRN) for providing HPC resources that have contributed to the research results
reported in this paper.

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
