# Peer review of "Reconstruction of Mercury's internal magnetic field beyond the octupole"

_Annales Geophysicae, 2021_

## Author Comment (AC1)

**Reviewer 1**

The manuscript "Reconstruction of Mercury's internal magnetic field beyond the octupole" by Toepfer et al. presents and compares different existing inversion techniques to reconstruct the internal magnetic field data of Mercury. The authors use a simulated magnetospheric model to get the synthetic magnetic field data for the inversions, to mimic the BepiColombo (MPO) mission data acquisition in a non current free environment. In this way, the authors are capable of evaluating the different methods in retrieving the known coefficients given a priori to the simulations. This study is of great importance specially that BepiColombo is on its way to Mercury. Besides some concerns detailed below, the manuscript is well organized and well written.

**Reply:** Thank you very much for reviewing the paper.

*Main comments:*

My main comment concerns the application to simulated Mercury magnetic field data. The authors make use of known Gauss coefficients from literature (Anderson et al. 2012, Thébault et al. 2018 and Wardinski et al. 2019), where the different models represent the Hermean internal magnetic field with Gauss coefficients of m=0. Those models are evidently constrained by the available data over a single hemisphere.

As the authors are doing a theoretical exercise using a synthetic model, I wonder if the authors shouldn't test the different inversion techniques for a more complex internal magnetic field, where m!=0 coefficients are also considered. For example, I have in mind the Jupiter's internal magnetic field case where the magnetic field from a hemisphere is axisymmetric but not in the other. In the case Mother Nature surprises us with a more exotic Hermean internal magnetic field, are the different inversion methods described in this manuscript capable of representing the non-axial Gauss coefficients? I suggest to run a couple of simulations using non-axisymmetric coefficients to test the limits of the different inversion techniques combined with the Gauss-Mie parametrization.

**Reply:** Agreed. We will perform additional simulation where the reconstruction of m!=0 coefficients is considered. The simulations are currently running on the high performance computer.

My second main comment is related to the effectiveness of the different techniques to retrieve the Gauss Coefficients. This issue is actually highlighted when the authors used two different values for coefficient $g^0_5$. Depending

*on the strength of this coefficient the authors show that the inversion method perform differently. It should be mentioned that the used models (Anderson et al. 2012, Thébault et al. 2018 and Wardinski et al. 2019) give Gauss coefficients with a strong covariance, because there are no data available in the Southern Hemisphere. This means that several sets of coefficients are still valid to represent the internal field. Given that, I wonder what is the impact in changing slightly the many coefficient values (as done for g^0_5). How well are the different inversion techniques retrieving the given coefficient, and how much that value should vary to notice that change? I suggest the authors to try other $g_1^0$, $g_2^0$, $g_3^0$, ... values not too different from those already used, and that can potentially be also a solution for Mercury's magnetic field, to check if there are limitations in discerning one from the other.*

**Reply:** Agreed. We will perform an additional simulation with slightly different axisymmetric internal Gauss coefficients. The simulation is currently running on the high performance computer.

*Moderate comments:*

*Introduction: I find that a description of the state of the art of the existing internal magnetic field models using MESSENGER data is missing. The lack of an introduction on this topic is enhanced later in section 4.1. I would also specify the limitations of those models, for example, the coefficients that are covarying.*

**Reply:** Agreed. We will add a description about the current knowledge of Mercury's internal magnetic field resulting from the MESSENGER data analysis.

*Lines 96 - 100 Please, add a sentence to describe how the thin shell approximation is affecting the results.*

**Reply:** Agreed.

*Section 4.2 After running the simulations but before selecting the data for inversions, is there a noise added to each data point? What is the error level? Is there more than a value considered? How it affects the inversions? There is a brief mention on this topic at lines 368-371 but it is not satisfying.*

**Reply:** Agreed. We will disturb the data synthetically and add the resulting errors of the coefficients.

*Lines 309 - 312 It might help display a figure with the spherical grid points used for the inversions. Please, indicate the grid resolution used. Also, how*

*much time shall we wait for BepiColombo to acquire enough data for your inversions?*

**Reply:** Agreed. We will add a figure where the synthetically generated data points are illustrated.

*Conclusion: An important aspect on internal field modeling is the time variation (or secular variation). This should also be mentioned here, even if this is not the scope of this manuscript.*

**Reply:** Agreed. We will add this aspect as an outlook.

*Minor comments:*

*Lines 14-17 there is a lack of citations.*

**Reply:** Agreed. We will add a citation.

*Line 26 Wardinski et al 2019 also estimate the size of the core.*

**Reply:** Agreed. We will add this aspect in the introduction section.

*Lines 34-36 There is also a disadvantage, the authors are modeling the external sources that are given by the simulations only, which could not be a full representation of the real magnetosphere currents.*

**Reply:** Agreed. We will clearly emphasize that the simulated data are a proxy for the not yet available MPO data.

*Lines 87 - 89 Please, define O in the text from equations 8 and 9.*

**Reply:** Agreed.

*Table 1 and 2: You have described in section 3.1 the Least Square Fit method, I would also add a column for that method in the given tables for comparison purposes.*

**Reply:** Making use of the truncated singular value decomposition, the Least Sqare Fit estimator transists into the TSVD estimator. Thus, comparing the LSF estimator with Capon's method (with truncated singular values) and the Tikhonov regularization may be declared as „unfair".

*Typos:*

*line 176 Imagenary -> Imaginary*

**Reply:** Agreed.

---

## Author Comment (AC2)

**Reviewer 2**

*The manuscript "Reconstruction of Mercury's internal magnetic field beyond the octupole" by S. Toepfer and co-authors provides an interesting comparison between different inversion methods to reconstruct the internal magnetic field of Mercury. They used a hybrid plasma code to simulate Mercury's magnetosphere, then they applied different methods to evaluate the "a priori" known coefficients. I think that the manuscript is a valuable contribution, especially for the BepiColombo community but not only. Indeed, these findings could be widely expanded to different environments, as well as, a similar methodological framework is a valuable support to any planetary mission. Furthermore, the manuscript is clear, well written, well organized, and well posed in terms of the existing literature. I would only suggest some improvements for the benefit of the reader as well as to assess their results.*

**Reply:** Thank you very much for reviewing the paper.

1. *I would recommend the authors to add errors on their coefficients' estimations as reported in Tables 1 and 2. Since both tables contains the main results of the paper, adding errors could improve the clarity of the results.*
   **Reply:** Agreed. We will disturb the data synthetically and add the resulting estimation errors.

2. *I would suggest the authors to comment on confidence intervals changes in their estimations under different solar wind conditions. Are they related to some specific solar wind parameters, apart the interplanetary magnetic field? I was wondering on plasma parameters like the Mach number(s) or the plasma beta. It would be nice to estimate coefficients under two/three different solar wind conditions or to add a few lines on this aspect.*
   **Reply:** Agreed. We will discuss this aspect within the summary.

3. *Another possible interesting aspect to be mentioned could be the role of considering different harmonics degrees in terms of both l and m. Could the authors comment on the expected changes as a function of m and l?*
   **Reply:** Agreed. We will perform additional simulation where the reconstruction of m!=0 coefficients is considered. The simulations are currently running on the high performance computer.

4. *I would suggest to add a few details on some specifics on the inversion models (noise, grid size, resolutions, ...) for clarity.*
   **Reply:** Agreed. We will add a figure where the grid size and resolution of the measurement points is illustrated.

5. *In light of the application of inversion methods to BepiColombo data, I would ask the authors to comment on the following aspect. The authors will use the model on magnetic field time series when MPO will explore different regions on the Hermean environment. I was wondering how the fast temporal variability of the Hermean environment as well as that of the different regions could affect the inversion methods used. Could this be considered as a "noise" for the method? Should be useful to firstly apply some filtering procedures on magnetic field data to remove the short-term variability and then apply the inversion method to the large-scale variability of MAG measurements? Could be some mixing between temporal and spatial scales that could affect the model performances?*

   **Reply:** Agreed. We will dicuss this aspect within the summary.

---

## Author Response (AR1)

**Reviewer 1**

*The manuscript "Reconstruction of Mercury's internal magnetic field beyond the octupole" by Toepfer et al. presents and compares different existing inversion techniques to reconstruct the internal magnetic field data of Mercury. The authors use a simulated magnetospheric model to get the synthetic magnetic field data for the inversions, to mimic the BepiColombo (MPO) mission data acquisition in a non current free environment. In this way, the authors are capable of evaluating the different methods in retrieving the known coefficients given a priori to the simulations. This study is of great importance specially that BepiColombo is on its way to Mercury. Besides some concerns detailed below, the manuscript is well organized and well written.*

**Reply:** Thank you very much for reviewing the paper.

*Main comments:*

*My main comment concerns the application to simulated Mercury magnetic field data. The authors make use of known Gauss coefficients from literature (Anderson et al. 2012, Thébault et al. 2018 and Wardinski et al. 2019), where the different models represent the Hermean internal magnetic field with Gauss coefficients of m=0. Those models are evidently constrained by the available data over a single hemisphere.*

*As the authors are doing a theoretical exercise using a synthetic model, I wonder if the authors shouldn't test the different inversion techniques for a more complex internal magnetic field, where m!=0 coefficients are also considered. For example, I have in mind the Jupiter's internal magnetic field case where the magnetic field from a hemisphere is axisymmetric but not in the other. In the case Mother Nature surprises us with a more exotic Hermean internal magnetic field, are the different inversion methods described in this manuscript capable of representing the non-axial Gauss coefficients? I suggest to run a couple of simulations using non-axisymmetric coefficients to test the limits of the different inversion techniques combined with the Gauss-Mie parametrization.*

**Reply:** Agreed. We performed an additional simulation where the reconstruction of m!=0 coefficients is considered (p. 21, ll. 421–434 and p. 23, Tab. 4).

*My second main comment is related to the effectiveness of the different techniques to retrieve the Gauss Coefficients. This issue is actually highlighted when the authors used two different values for coefficient $g^0_5$. Depending*

*on the strength of this coefficient the authors show that the inversion method perform differently. It should be mentioned that the used models (Anderson et al. 2012, Thébault et al. 2018 and Wardinski et al. 2019) give Gauss coefficients with a strong covariance, because there are no data available in the Southern Hemisphere. This means that several sets of coefficients are still valid to represent the internal field. Given that, I wonder what is the impact in changing slightly the many coefficient values (as done for g^0_5). How well are the different inversion techniques retrieving the given coefficient, and how much that value should vary to notice that change? I suggest the authors to try other g_1^0, g_2^0, g_3^0, … values not too different from those already used, and that can potentially be also a solution for Mercury's magnetic field, to check if there are limitations in discerning one from the other.*

**Reply:** Agreed. We performed an additional simulation with slightly different axisymmetric internal Gauss coefficients and discussed the results in analogy to the varying value of g50 (p. 21, ll. 403–420 and p. 22, Tab. 3).

*Moderate comments:*

*Introduction: I find that a description of the state of the art of the existing internal magnetic field models using MESSENGER data is missing. The lack of an introduction on this topic is enhanced later in section 4.1. I would also specify the limitations of those models, for example, the coefficients that are covarying.*

**Reply:** Agreed. We added a description about the current knowledge of Mercury's internal magnetic field resulting from the MESSENGER data analysis and specified the limitations (p. 2, ll. 35–42).

*Lines 96 - 100 Please, add a sentence to describe how the thin shell approximation is affecting the results.*

**Reply:** Agreed. We described how the thin shell approximation affects the results (p. 4, ll. 111–112).

*Section 4.2 After running the simulations but before selecting the data for inversions, is there a noise added to each data point? What is the error level? Is there more than a value considered? How it affects the inversions? There is a brief mention on this topic at lines 368-371 but it is not satisfying.*

**Reply:** Agreed. We disturbed the data synthetically and added the resulting errors of the coefficients (p. 17, Tab. 1 and p. 18, ll. 369–384).

*Lines 309 - 312 It might help display a figure with the spherical grid points used for the inversions. Please, indicate the grid resolution used. Also, how much time shall we wait for BepiColombo to acquire enough data for your inversions?*

**Reply:** Agreed. We added a figure where the synthetically generated data points are illustrated (p. 16, Fig. 2).

*Conclusion: An important aspect on internal field modeling is the time variation (or secular variation). This should also be mentioned here, even if this is not the scope of this manuscript.*

**Reply:** Agreed. We added this aspect on p. 25, ll. 483–485 as an outlook.

*Minor comments:*

*Lines 14-17 there is a lack of citations.*

**Reply:** Agreed. We added proper citations (p. 1, l. 14 and l. 17).

*Line 26 Wardinski et al 2019 also estimate the size of the core.*

**Reply:** Agreed. We added this aspect in the introduction section (p. 2, ll. 25–26).

*Lines 34-36 There is also a disadvantage, the authors are modeling the external sources that are given by the simulations only, which could not be a full representation of the real magnetosphere currents.*

**Reply:** Agreed. We clearly emphasized that the simulated data are a proxy for the not yet available MPO data (p. 2, l. 46).

*Lines 87 - 89 Please, define O in the text from equations 8 and 9.*

**Reply:** Agreed. We added the definition of the Big-O-notation (p. 4, l. 104).

*Table 1 and 2: You have described in section 3.1 the Least Square Fit method, I would also add a column for that method in the given tables for comparison purposes.*

**Reply:** Making use of the truncated singular value decomposition, the Least Square Fit estimator transists into the TSVD estimator. Thus, comparing the LSF estimator with Capon's method (with truncated singular values) and the Tikhonov regularization may be declared as „unfair".

*Typos:*

*line 176 Imagenary -> Imaginary*

**Reply:** Agreed.

**Reviewer 2**

*The manuscript "Reconstruction of Mercury's internal magnetic field beyond the octupole" by S. Toepfer and co-authors provides an interesting comparison between different inversion methods to reconstruct the internal magnetic field of Mercury. They used a hybrid plasma code to simulate Mercury's magnetosphere, then they applied different methods to evaluate the "a priori" known coefficients. I think that the manuscript is a valuable contribution, especially for the BepiColombo community but not only. Indeed, these findings could be widely expanded to different environments, as well as, a similar methodological framework is a valuable support to any planetary mission. Furthermore, the manuscript is clear, well written, well organized, and well posed in terms of the existing literature. I would only suggest some improvements for the benefit of the reader as well as to assess their results.*

**Reply:** Thank you very much for reviewing the paper.

1. *I would recommend the authors to add errors on their coefficients' estimations as reported in Tables 1 and 2. Since both tables contains the main results of the paper, adding errors could improve the clarity of the results.*
   **Reply:** Agreed. We disturbed the data synthetically and added the resulting estimation errors (p. 17, Tab. 1 and p. 18, ll. 369–384).

2. *I would suggest the authors to comment on confidence intervals changes in their estimations under different solar wind conditions. Are they related to some specific solar wind parameters, apart the interplanetary magnetic field? I was wondering on plasma parameters like the Mach number(s) or the plasma beta. It would be nice to estimate coefficients under two/three different solar wind conditions or to add a few lines on this aspect.*
   **Reply:** Agreed. We discussed this aspect on p. 25, ll. 476–481 within the summary.

3. *Another possible interesting aspect to be mentioned could be the role of considering different harmonics degrees in terms of both l and m. Could the authors comment on the expected changes as a function of m and l?*
   **Reply:** Agreed. We performed an additional simulation where the reconstruction of m!=0 coefficients is considered (p. 21, ll. 421–434 and p. 23, Tab. 4).

4. *I would suggest to add a few details on some specifics on the inversion models (noise, grid size, resolutions, ...) for clarity.*
   **Reply:** Agreed. We added a figure where the grid size and resolution of the measurement points is illustrated (p. 16, Fig. 2).

5. *In light of the application of inversion methods to BepiColombo data, I would ask the authors to comment on the following aspect. The authors will use the model on magnetic field time series when MPO will explore different regions on the Hermean environment. I was wondering how the fast temporal variability of the Hermean environment as well as that of the different regions could affect the inversion methods used. Could this be considered as a "noise" for the method? Should be useful to firstly apply some filtering procedures on magnetic field data to remove the short-term variability and then apply the inversion method to the large-scale variability of MAG measurements? Could be some mixing between temporal and spatial scales that could affect the model performances?*

**Reply:** Agreed. We dicussed this aspect on p. 25, ll. 476–481 within the summary.

**General changes in the manuscript:**

- Changes in the manuscript are marked with „latexdiff", i.e., added text is marked in blue and the old version of the formulation is crossed out and marked in red.
- The position of changes that are related to Reviewer comments are directly stated in the reply.
- p. 1, l. 14 and l. 17: We added proper citations.
- p. 2, ll. 35–42:  We added a description about the current knowledge of Mercury's internal magnetic field resulting from the MESSENGER data analysis.
- p. 2, l. 46: We clearly emphasized that the simulated data are a proxy for the not yet available in-situ data.
- p. 4, l. 104: We explained the Big-O-notation.
- p. 4, ll. 111–112: We explained how the thin shell approximation improves the inversion results.
- p. 16: We added a figure where the distribution of the synthetically generated data points is illustrated.
- p. 17, Tab. 1: We synthetically disturbed the measurements and the measurement positions and added the resulting estimation error.
- p. 18, ll. 369–384: We explained how the  synthetically generated noise is constructed.
- p. 21, ll. 403–420 and Tab. 3: We performed an additional simulation to reconstruct the axisymmeric internal Gauss coefficients with slightely different numerical values.
- p. 21, ll. 421–434 and Tab. 4: We performed an additional simulation to reconstruct potentially existing nonaxisymmeric internal Gauss coefficients.
- p. 24, ll. 458–460 and ll. 462–464: We discussed the results of the additionally performed simulations within the summary.
- p. 25, ll. 476–481: We mentioned the influence of temporal changes within Mercury's magnetosphere as well as different solar wind conditions.
- p. 25, ll. 483–485: We mentioned the ability for analyzing potentially existing secular variations of Mercury's internal magnetic field.
- p. 28–29: The literature list has been extended by the papers of Narita et al. (2021), Oliveira et al. (2019), Philpott et al. (2014), Slavin et al. (2021) and Wang et al. (2012)

---

## Author Response (AR2)

**Reviewer 1**

*The second version of "Reconstruction of Mercury's internal magnetic field beyond the octupole" from Toepfer et al. manuscript tackles well the comments raised by the reviewers from a previous version of the manuscript. For the sake of completeness/clearness of the study, I have few minor comments for the authors to consider.*

*Note: I am following the manuscript version with tracked changes for the line numbering.*

**Minor comments:**

*l 432 - I am not sure if I agree with the statement, specifically with "good precision". In my opinion the authors should develop what it means in this case. When an axisymmetric Gauss coefficient (large coefficient values) is retrieved with only 1 to 2 nT difference, it corresponds to a low % of the initial Gauss coefficient value. However, 1 or 2 nT difference in retrieving non-axisymmetric coefficients (usually lower values) is a large % of the initial value, and therefore the Gauss coefficient is retrieved with possibly not "good precision" but rather "fair precision". In order to avoid misunderstanding in reading this sentence, I would suggest the authors to add relative estimation errors to non-axisymmetric coefficients only, axisymmetric coefficients only, besides all coefficients (already given by the authors). It might happen that the performance of the modeling is not similar between axisymmetric and non-axisymmetric coefficients.*
*Also, what is the reason of not trying to have a g^4_1and g^5_1 != 0 in the new tests? This should be mentioned in the text.*

**Reply:** Agreed. We discussed the errors separately and mentioned the ability for testing different combinations and numerical values of nonaxisymmetric internal Gauss coefficients in future studies (p. 22, ll. 435–445).

*l 35 - 43 - I find the paragraph can be further enriched with introductory content as the 3 manuscripts cited here are not the only ones with internal field models using MESSENGER data. The authors can refer to Heyner et al. 2021 for a complete review of internal field models state of the art, however 2 new papers have been published since then: Wardinski et al 2021 and Plattner and Johnson 2021 (I note however, that the authors could not cite these two papers in a previous version of the manuscript because they are very recent publications).*
*I would suggest to add few lines to this paragraph making a more complete review of the various existing internal field models, and the corresponding modeling aspects. Details missing are: 1) what are the general techniques used by the different authors? Is there a technique more robust than the others? 2) what data each of them are using (a 2012 publication cannot be using all the spacecraft data, for example). 3) are there corrections to the external fields or how the authors are separating internal from external source contributions?*

*All these details strongly impacts the Gauss coefficients later used by the authors.*

**Reply:** Agreed. We extended the introduction section and the literature list accordingly (p. 2, ll. 35–38, ll. 43–46).

*l481 I would suggest the authors to update the Benkhoff et al. 2010 citation to Benkhoff et al. 2021.*

**Reply:** Agreed.

***Other:***

*l 419-420 - Probably this comment is out of the scope of this work, but I'll leave it here for future consideration from the authors. This sentence raises the question of: for what coefficient value the performance of the different methods start to clearly decrease?*

**Reply:** This is a very interesting question that should be discussed by performing further simulation and parameter studies in future works.

*Typos:*

*l37 "describes" -> "described"*

**Reply:** Agreed.

*l485 "Olveira" -> "Oliveira"*

**Reply:** Agreed.

**General changes in the manuscript**

- Changes in the manuscript are marked with „latexdiff", i.e., added text is marked in blue and the old version of the formulation is crossed out and marked in red.
- The position of changes that are related to Reviewer comments are directly stated in the reply.
- p. 2, ll. 35–38, ll. 43–46: The introduction section has been extended by the existing models of Mercury's magnetic field.
- p. 22, ll. 435–445: The relative reconstruction errors for the axisymmetric and the nonaxisymmetric internal Gauss coefficients are discussed separately.
- p. 28–30: The literature list has been extended by the papers of Benkhoff et al. (2021), Oliveira et al. (2015), Plattner and Johnson (2021) and Wardinski et al. (2021).